# Identifying Risk Factors for Hospitalization with Behavioral Health Disorders and Concurrent Temperature-Related Illness in New York State

**DOI:** 10.3390/ijerph192416411

**Published:** 2022-12-07

**Authors:** Heather Aydin-Ghormoz, Temilayo Adeyeye, Neil Muscatiello, Seema Nayak, Sanghamitra Savadatti, Tabassum Z. Insaf

**Affiliations:** 1Center for Environmental Health, New York State Department of Health, Albany, NY 12208, USA; 2School of Public Health, University at Albany, Rensselaer, NY 12144, USA

**Keywords:** mental illness, heat-related illness, cold-related illness, behavioral health disorder hospitalizations

## Abstract

Extreme temperature events are linked to increased emergency department visits, hospitalizations, and mortality for individuals with behavioral health disorders (BHD). This study aims to characterize risk factors for concurrent temperature-related illness among BHD hospitalizations in New York State. Using data from the NYS Statewide and Planning Research and Cooperative System between 2005–2019, multivariate log binomial regression models were used in a population of BHD hospitalizations to estimate risk ratios (RR) for a concurrent heat-related (HRI) or cold-related illness (CRI). Dementia (RR 1.65; 95% CI:1.49, 1.83) and schizophrenia (RR 1.38; 95% CI:1.19, 1.60) were associated with an increased risk for HRI among BHD hospitalizations, while alcohol dependence (RR 2.10; 95% CI:1.99, 2.22), dementia (RR 1.52; 95% CI:1.44, 1.60), schizophrenia (RR 1.41; 95% CI:1.31, 1.52), and non-dependent drug/alcohol use (RR 1.20; 95% CI:1.15, 1.26) were associated with an increased risk of CRI among BHD hospitalizations. Risk factors for concurrent HRI among BHD hospitalizations include increasing age, male gender, non-Hispanic Black race, and medium hospital size. Risk factors for concurrent CRI among BHD hospitalizations include increasing age, male gender, non-Hispanic Black race, insurance payor, the presence of respiratory disease, and rural hospital location. This study adds to the literature by identifying dementia, schizophrenia, substance-use disorders, including alcohol dependence and non-dependent substance-use, and other sociodemographic factors as risk factors for a concurrent CRI in BHD hospitalizations.

## 1. Introduction

Average annual temperatures in New York State (NYS) are expected to rise 2.2–5 °C by 2080 and the state can expect increased days of extreme heat in the warmer months over the next 60 years [1]. Most NYS regions are projected to have an average of 21–74 days over 32 °C by 2080 [1]. Despite this, NYS remains a region that experiences extreme cold events in the winter months, although these events are expected to decrease given the predicted rise in average temperatures [1]. By 2080, it is projected that NYS will have an average of 67–116 days at or below 0 °C [1]. NYS experiences large variations in regional temperatures and no single definition can be used to define extreme weather events across the state. In the United States from 2006 to 2010, 3340 people died from heat-related illness (HRI) and 6652 people died from cold-related illness (CRI) [2]. While previous studies also suggest that most temperature-associated mortality is a result of CRI, increasing ambient temperatures make mortality from HRI an emerging threat in NYS, especially for vulnerable populations [3].

Extreme heat and cold temperatures can impact health directly as well as exacerbate existing medical conditions and, in some instances, lead to death [3,4,5,6,7]. Prolonged exposure to high temperatures causes HRI, including heat stress, heat stroke and heat exhaustion, while prolonged exposure to low temperatures causes a variety of CRI, including hypothermia and frostbite. Both exposures can exacerbate medical conditions, such as respiratory and cardiovascular diseases. Several research studies from around the world have found associations between temperature increases or heatwaves and increased mental health emergency department (ED) visits and hospitalizations [8,9,10,11,12,13,14,15]. ED visits for behavioral health disorders, schizophrenia, mood disorders, neurotic disorders, and morbidity due to substance-use have also been shown to be associated with high temperature [16]. High ambient temperatures are also associated with increased mortality in populations with behavioral disorders, specifically substance-use disorders [17,18,19]. Behavioral health disorders (BHD) have also been found to be significant risk factors of HRI [11]. A study conducted in the U.S. in 2017 by Schmeltz and Gamble analyzed specific BHD diagnoses and the risk of HRI hospitalizations, as well as individual and environmental risk factors. Schmeltz and Gamble found that among BHD hospitalizations, the risk of a concurrent HRI was higher among patients with dementia, psychoactive substance abuse, and nondependent drug abuse [11]. Schmeltz and Gamble (2017) also analyzed risk factors by patient characteristics and showed an increased risk among African Americans, Native Americans, males, and adults over the age of 40 [11]. Additionally, those with lower income, the uninsured, rural patients, and those going to smaller hospitals had an increased risk of hospitalization for BHD with concurrent HRI [11].

The association between low ambient temperature and BHD-related ED visits, hospitalizations, and mortality is inconclusive in the literature. Research finds that mental health comorbid conditions are common in CRI mortality [20]. Additionally, antipsychotic medications, in the presence of multiple risk factors, have been observed to cause hypothermia in the first 7–10 days of initiation or when increasing the dosage [21]. Other research has suggested that cold temperatures are found to decrease adverse mental health outcomes [22,23].

There are a limited number of studies that have investigated, in a population of people hospitalized with BHD, which risk factors contributed to concurrent temperature-related illness (TRI). Due to climate, temperature, and demographic variation between US regions, there is also a need to conduct this study in NYS to identify state-specific risk factors. In this study, we aim to determine in a population of BHD hospitalizations which BHD have an increased risk of concurrent TRI in NYS. Furthermore, we seek to identify the demographic factors associated with an increased risk of a concurrent cold- or heat-related illnesses in NYS in the same population.

## 2. Materials and Methods

This study uses cross-sectional administrative data from 2005–2019 to identify possible risk factors in the population of BHD hospitalizations for TRIs. The primary data source for this study was hospitalization data from the NYS Statewide Planning and Research Cooperative System (SPARCS). NYS SPARCS provides several categories of statewide patient information data, including demographic data, such as age, sex, race, and ethnicity, insurance payor data, primary and secondary diagnoses, and hospital information for all hospital admissions in NYS excluding Federal and Veterans Health Administration hospitals, Indian Health Service hospitals, and institutionalized (e.g., prison) populations. The risk factors considered in this analysis were specific BHD diagnoses, including age, race, sex, insurance status, comorbidities, hospital size, and hospital location. Hospital size and location were included to assess risk in rural communities where mental health services are less accessible. Variables selected for analysis were based on prior U.S. national research on risk factors for BHD with concurrent HRI, specifically the research conducted by Schmeltz and Gamble [11]. This research was chosen as it provided national level estimates of risk factors for hospitalization with BHD and HRI that were useful to compare to findings in NYS.

### 2.1. Population

We analyzed administrative claims data of patients aged five and over who had been hospitalized in NYS between 2005 and 2019 with a primary or secondary diagnosis of BHD. We excluded BHD hospitalization under the age of five since diagnosis of BHD in very young children is difficult and potentially inaccurate. The classification of mental illness for this analysis was developed based on a review of the prior literature and expert guidance from the NYS Office of Mental Health.

This study includes a full analysis and two sub-analyses of risk factors for concurrent TRI in BHD hospitalizations. The full BHD analysis includes all mental disorder diagnoses in addition to dementia and the two sub-analyses are limited to mental disorder diagnoses (excluding dementia) and substance-use disorders diagnoses. The mental disorder categories for this study included depressive disorders, schizophrenia spectrum and other psychotic disorders, bipolar and related disorders, anxiety disorders, trauma and stressor related disorders, attention deficit hyperactivity disorder, disruptive disorders, impulse control disorders, conduct disorders, personality disorders, and obsessive compulsive and related disorders. The International Classification of Diseases, Ninth Revision (ICD-9) to Tenth Revision (ICD-10) crosswalk was developed using the Center for Medicare and Medicaid Services’ General Equivalence Mappings (see Appendix A for the BHD Crosswalk) [24]. We included dementia in the full analysis, despite it often being considered a neurological disorder, as previous research demonstrated an association between hospitalization for dementia and concurrent HRI [11]. Additionally, cooler than average temperatures and high temperature variability have been linked to increased dementia hospitalizations [25]. The substance-use disorders categories include alcohol dependence, drug dependence, and non-dependent drug and alcohol use (see Appendix A for the BHD Crosswalk).

### 2.2. Outcomes

The two main outcomes of interest in this study were hospitalization with BHD and a concurrent HRI diagnosis during the warmer months (May-September), and hospitalization with BHD and a concurrent CRI during the colder months (October–April). TRI data were obtained using hospital administrative coding for HRI and CRI from the NYS SPARCS dataset. Among hospitalizations with BHD and a concurrent HRI diagnosis, HRI was classified as heat stroke, sunstroke, heat syncope, heat exhaustion and other heat effects, accidents caused by excessive heat due to weather conditions, and exposure to excessive natural heat and sunlight (ICD-9: 992, E900.0, E900.9; ICD-10: T67, X30, X32). Approximately 95% of HRI diagnoses were made between May and September. Patients hospitalized with BHD and a concurrent HRI were compared to patients hospitalized with only BHD and no HRI during the warmer months. Among hospitalizations with BHD and a concurrent CRI diagnosis, CRI was classified as hypothermia, frostbite with tissue necrosis, trench foot, chilblains, other effects of reduced temperature, accident due to excessive cold due to weather conditions, and injury by extremes of cold (ICD-9: 991, E901.0, E901.9, E988.3; ICD-10: T68, T69, X31, T33, T34). Approximately 82% of CRI diagnoses were made between October and April. Patients hospitalized with BHD and a concurrent CRI were compared to patients hospitalized with only BHD and no CRI during the colder months.

### 2.3. Covariates

#### 2.3.1. Individual Level Covariates

These included demographics (age, sex, and race/ethnicity), health insurance and concurrent comorbidities. We categorized insurance payor into Medicare, Medicaid, private/HMO, uninsured, or other (Worker’s Compensation, Civilian Health, and Medical Program of the Uniformed Service (CHAMPUS), Title V, other government programs, and non-government programs). Comorbidities for TRIs include diabetes, cardiovascular disease, respiratory disease, and obesity (See Appendix A for comorbidities crosswalk) [20,26,27].

#### 2.3.2. Hospital and Area Level Covariates

The hospital size variable describes urbanicity and health resources. It was developed for this study using a classification from the Healthcare Cost and Utilization Project (HCUP) Statistical Brief using number of hospital beds, location, and teaching status to determine size [28]. The hospital location variable describes urbanicity of the census tract it lies within and was developed using the Rural-Urban Commuting Area (RUCA) codes—Classification B. RUCA codes provide a classification of U.S. census tracts using measures of population density, urbanization, and daily commuting [29]. The most recent RUCA codes in use are based on the 2010 decennial census and the 2006–2010 American Community Survey. Schmeltz and Gamble included an income variable in their analysis using Census tract income quartiles based on residential ZIP codes. We did not create an income variable from Census tract income quartiles since patients with address information missing in SPARCS would have been excluded from complete case analysis. Given the high prevalence of BHD and exposure to extreme temperatures in homeless populations, the exclusion of patients with no addresses would not be useful to this analysis. SPARCS does include a homelessness variable; however, it was considered unreliable since there was a significant difference between the ICD-9 (5.9%) and ICD-10 (0.4%) classification years (*p* < 0.0001), and the prevalence of homelessness in the BHD population is expected to be higher.

### 2.4. Statistical Analysis

A univariate analysis was performed on all individual and hospital level variables; frequencies and distributions of all relevant variables were assessed for all BHD hospitalizations, for BHD with concurrent CRI, and for BHD with concurrent HRI. Subsequently, a bivariate analysis using the Chi-squared test was conducted to assess demographic differences between hospitalizations with and without the outcome of interest. Frequency counts and percentages of BHD with concurrent CRI and HRI were also conducted. A bivariate log-binomial regression was conducted to estimate relative risks (RR) of specific mental health diagnoses for BHD hospitalization with a concurrent CRI or HRI [30,31,32,33]. Cross-sectional research with a binary outcome such as this study often uses logistic regression; however, using log-binomial regression avoids overestimation of the prevalence odds ratio by directly estimating the prevalence ratio [30]. Where BHD counts were less than 15, we ran exact logistic regressions to confirm the findings. The results were very similar to the log-binomial RRs, with incrementally tighter precision. Subsequently, multivariable log-binomial regression was conducted using predictive models and complete case analysis to assess the individual and regional risk factors for hospitalization with BHD and a concurrent TRI.
log(πTRI) = β0 + β1 Age + β2 Sex + β3 Race + β4 Payor + β5 Comorbid Conditions + β6 Hospital Size + β7 Hospital Location (1)

Statistical analyses for this study were performed using SAS software (SAS 9.4, SAS Institute, Cary, NC, USA).

## 3. Results

NYS is located in the Northeastern part of the United States and experiences wide variations in regional temperatures. The temperature distribution during the warm (May through September) and cold months (October through April) of the study period (2005–2019) varied widely (Appendix A). In summary, the average maximum temperature for the warm months in NYS was 24.72 °C (76.50 °F) while the cold months was 7.99 °C (46.38 °F). The range of maximum temperature during the study period was from −0.67 °C to 39.21 °C (30.79 °F to 102.58 °F) for warm months and −25.06 °C (−13.11 °F) through 31.63 °C (88.93 °F) for the cold months. The average maximum heat index during the warm months was 25.53 °C (77.95 °F) with a range of −0.67 °C to 50.28 °C (30.79 °F to 122.50 °F), while during the cold months, the maximum heat index ranged from −25.06 °C to 34.67 °C (−13.11 °F to 94.41 °F), with an average maximum heat index of 7.46 °C (45.43 °F).

The study population consists of 12,272,393 BHD hospitalizations from 2005 to 2019 among NYS residents, including those of New York City. There was a total of 18,872,852 diagnoses of the selected BHD in our analysis. About 62% of hospitalizations had one BHD diagnosis, 22% had two BHD diagnoses, and 11% had three diagnoses (Appendix A). Table 1 provides a summary of the demographic distribution of the study population. The mean age in the BHD hospitalizations study population was 56 years, with about 41% of hospitalizations in the 40–64 age category. About 51% of the study population were female. The race and ethnicities of the study population were non-Hispanic white (57%), Hispanic/Latino (12%), non-Hispanic Black (19%), Asian/Pacific Islander (2%), and other (11%); other races included Native American, Non-Hispanic multiracial, and other races. The insurance payor status in the study population was Medicare (35%), Medicaid (37%), private insurance (19%), and uninsured (8%). The comorbidity frequency was diabetes (23%), cardiovascular disease (22%), respiratory disease (35%), and obesity (10%). The majority of hospitalizations were in large hospitals (54%) or in urban locations (92%) (Table 1).

Among BHD hospitalizations, significantly decreased risks of concurrent TRI were observed for several diagnoses including depressive disorders, anxiety disorders, and bipolar disorders compared to BHD hospitalizations without these diagnoses (Table 2). Multivariate log-binomial models characterizing the individual and regional risk factors for a concurrent TRI among BHD hospitalizations are detailed in Table 3 and Table 4.

### 3.1. BHD Hospitalizations with Concurrent HRIs

There were 1914 BHD hospitalizations with concurrent HRIs from 2005–2019 during the months of May through September; and this varied between 50–200 cases per year and did not exhibit a discernible trend over time during the study period (Appendix A). The characteristics of the BHD hospitalizations with concurrent HRI are summarized in Table 1 and were significantly different from those with BHD alone for several variables including age, sex, race, insurance payer, and hospital size (*p* < 0.0001–0.01), (Table 1). Less than 2% of BHD hospitalizations with HRI and a comorbidity occurred in the same individual.

In Table 2, we examine the frequency and distribution of the selected BHD with concurrent TRI diagnoses. Of the 1914 BHD hospitalizations with a concurrent HRI, there were 2661 BHD diagnoses. Mental illness accounted for 48% of BHD with concurrent HRI diagnoses, including depressive disorders (18%), schizophrenia spectrum and other psychotic disorders (13%), and anxiety disorders (8%). Dementia accounted for 18% of BHD with concurrent HRI diagnoses. The other 34% of the diagnoses were substance-use disorders; of which the majority was accounted for by non-dependent drug use (70%). Log-binomial regression analysis showed that dementia (RR 1.65; 95% Confidence Interval (CI): 1.49, 1.83) and schizophrenia spectrum and other psychotic disorders (RR 1.24; 95% CI: 1.10, 1.39), including schizophrenia (RR 1.38; 95% CI:1.19, 1.60), were associated with an increased risk for HRI among BHD hospitalizations. In a sub-analysis of multiple diagnoses among BHD hospitalizations, we did not observe an increased risk of HRI in those with dementia and schizophrenia, dementia and substance-use disorders, and schizophrenia and substance-use disorders compared to those without these multiple diagnoses (Appendix A).

Risk factors for hospitalization for BHD and concurrent HRI include age, specifically 40–64 years (RR 1.46; 95% CI: 1.27, 1.69), 65–74 years (RR 2.26; 95% CI: 1.87, 2.73), and 75 years and over (RR 2.62; 95% CI: 2.19, 3.13), male gender (RR 1.87; 95% CI: 1.69, 2.06), non-Hispanic Black race (RR 1.34; 95% CI: 1.18, 1.51) and medium hospital size (RR 1.18; 95% CI: 1.07, 1.32). Significantly decreased risks of hospitalization for MBD and concurrent HRI were associated with cardiovascular disease (RR 0.83; 95% CI: 0.74, 0.93) and respiratory disease (RR 0.86 95% CI: 0.78, 0.95). In the sub-analysis of hospitalization with a mental disorder and concurrent HRI, older age and non-Hispanic Black persisted as risk factors. In a sub-analysis of hospitalization with substance-use disorders and concurrent HRI, older age and male gender persisted as risk factors. Additionally, being uninsured, having other insurance status, and the presence of respiratory disease were found to be risk factors. A significantly decreased risk of hospitalization for substance-use disorders and concurrent HRI was associated with cardiovascular disease (RR 0.80; 95% CI: 0.66, 0.97) (Table 3).

### 3.2. BHD Hospitalizations with Concurrent CRIs

There were 7738 BHD hospitalizations with concurrent CRIs from 2005–2019 during the months of October to April. BHD hospitalizations with concurrent CRI diagnoses trended upward over the study period, increasing from an average of 400 to over 1000 per year, or from 5% to 11% (Appendix A). The characteristics of those hospitalized with concurrent CRI are also summarized in Table 1 and were significantly different from those with BHD alone for all variables (*p* < 0.0001–0.05). In BHD hospitalizations, 12% with CRI and diabetes, 8% with CRI and respiratory disease, 7% with CRI and CVD, and 2% with CRI and obesity occurred in the same individual with two to three hospitalizations over the study period.

Of the 7738 hospitalizations of BHD with a concurrent CRI, there were 11,449 BHD diagnoses. Mental illness accounted for 44% of BHD with concurrent CRI diagnoses, including depressive disorders (15%), schizophrenia spectrum and other psychotic disorders (12%), and anxiety disorders (7%). Dementia accounted for 16% of BHD with concurrent CRI diagnoses. The other 40% of the diagnoses were substance-use disorders, primarily non-dependent drug use (56%). A log-binomial regression analysis showed that dementia (RR 1.52; 95% CI:1.44, 1.60), schizophrenia spectrum and other psychotic disorders (RR 1.22; 95% CI: 1.15, 1.30), which included schizophrenia (RR 1.41; 95% CI:1.31, 1.52) were associated with an increased risk of CRI among BHD hospitalizations. Substance-use disorders (RR 1.62; 95% CI:1.55, 1.70), which included alcohol dependence (RR 2.10; 95% CI:1.99, 2.22), and non-dependent drug/alcohol use (RR 1.20; 95% CI:1.15, 1.26) were also associated with an increased risk of CRI among BHD hospitalizations. In a sub-analysis of multiple diagnoses among BHD hospitalizations, we observed an increased risk of concurrent CRI in those with dementia and schizophrenia, dementia and substance-use disorders, and schizophrenia and substance-use disorders compared to those without these multiple diagnoses (Appendix A).

Risk factors for hospitalization for BHD and concurrent CRI include age, specifically 40–64 years (RR 1.53; 95% CI: 1.43, 1.64), 65–74 years (RR 1.55; 95% CI: 1.40, 1.71), and 75 years and over (RR 1.78; 95% CI: 1.63, 1.95), male gender (RR 2.06; 95% CI: 1.96, 2.17), non-Hispanic Black race (RR 1.23; 95% CI: 1.15, 1.30), being on Medicare (RR 1.51; 95% CI: 1.38, 1.64), Medicaid (RR 1.71; 95% CI:1.59, 1.85), or uninsured status (RR 1.76; 95% CI: 1.59, 1.94), the presence of respiratory disease (RR 1.08; 95% CI: 1.03, 1.14), and hospitalization in a rural location, including a large rural city or town (RR 1.24; 95% CI: 1.13,1.36), or a small rural town (RR 1.54; 95% CI:1.35, 1.75). Significantly decreased risks of hospitalization for BHD and concurrent CRI were associated with diabetes (RR 0.89; 95% CI 0.84, 0.95), cardiovascular disease (RR 0.68; 95% CI: 0.64, 0.72) and obesity (RR 0.49; 95% CI: 0.43, 0.54). Sub-analysis of the risk factors for hospitalization with a mental disorder and concurrent CRI showed similar risk associations to the full model. Sub-analysis of risk factors for hospitalization with substance-use disorders and concurrent CRI included the following risk factors: being aged 5–17 years and increasing age, being on Medicare, Medicaid, or having uninsured status, and the presence of respiratory disease. Significantly decreased risks of hospitalization for substance-use disorders and concurrent CRI were associated with diabetes, cardiovascular disease, and obesity (Table 4).

## 4. Discussion

### 4.1. Mental Health Disorders

In the population of BHD hospitalizations in NYS, several mental health diagnoses and dementia were found to be risk factors for concurrent TRI. BHD hospitalized patients with dementia diagnoses were observed to be more likely to have a concurrent TRI compared to those without dementia diagnoses. BHD hospitalized patients with schizophrenia spectrum diagnoses and other psychotic disorders, particularly schizophrenia, were observed to be more likely to have a concurrent TRI compared to those without these diagnoses. Similar to findings in this study, Schmeltz and Gamble found dementia to be a risk factor for HRI [11]. People who have dementia or schizophrenia can have impaired decision making, communication, and comprehension. These factors may contribute to failure to stay adequately hydrated in extreme heat events, to dress appropriately for the weather, to seek shelter, to recognize the signs of TRI, and be vulnerable to prolonged exposure to extreme weather conditions. Dementia most commonly presents in older populations who are at increased risk of hyperthermia, hypothermia, and other TRI due to age related processes such as decreased metabolic rate, diminished sweating capacity, and inability to thermoregulate [34,35]. Additionally, hyperthermia and hypothermia are both known side-effects of antipsychotic medication used in the treatment of schizophrenia and dementia due to thermoregulatory disturbance [21,36,37].

### 4.2. Substance Abuse Disorders

Among BHD hospitalizations, patients with substance-use disorder diagnoses, alcohol dependence and non-dependent drug or alcohol use were observed to be more likely to have a concurrent CRI compared to patients without these diagnoses. Alcohol and drugs are known to impair mental state [35], behavioral responses, and can increase the risk of cold exposure [35]. Several substances, including alcohol, cocaine, and other stimulants have been found to affect thermoregulation [38,39].

### 4.3. Demographics

Among those hospitalized for BHD, older age groups, males, and non-Hispanic Blacks were associated with increased risk for a concurrent HRI or CRI. The HRI findings support those observed by Schmeltz and Gamble in their national research on risk factors for BHD with concurrent HRI [11]. Older age is a known risk factor for TRI, due to thermoregulatory deterioration and existing morbidities [34,35,40]. Older age remained a risk factor for concurrent TRI in the mental illness sub-analysis and for concurrent CRI in the substance-use disorders sub-analyses. Males with BHD hospitalization were observed to have an increased risk of concurrent TRI compared to females with BHD hospitalization; and the risk remained in the sub-analyses of cases diagnosed with mental illness and substance-use disorders. Males have previously been found to be at higher risk of hyperthermia and hypothermia compared to females [41,42]. The disparity in TRI risk may be attributed to a difference in risk-taking behavior, substance-use, occupational exposure, and utilization of healthcare services [11,43,44].

In the study population, having an additional concurrent diagnosis of respiratory disease was another risk factor for concurrent CRI compared to those BHD hospitalized without respiratory disease, and the risk remained in sub-analyses of both mental and substance-use disorders. Patients hospitalized with substance-use disorders had an increased risk of concurrent HRI if they had a concurrent respiratory infection. Respiratory diseases are found to be more prevalent in homeless populations and highly transmissible in homeless shelters [45,46]. Previous research also supports an association between exposure to extreme cold and increased risk of respiratory infection [20,47]. Unfortunately, in our analysis, data on the housing status of patients was not available, but future analyses on this risk factor relative to housing status is warranted. Protective effects for hospitalization with BHD and concurrent TRI associated with comorbidities such as diabetes, cardiovascular disease, and obesity need to be explored further, but may be due to the fact that those who are sick with multiple comorbidities have a lower risk of exposure, as they may not be outdoors for long periods of time. The observed decreased risk could also be due to the underreporting of secondary diagnoses, where comorbidity diagnoses take precedence in treatment and are recorded and reported instead of TRI diagnoses.

In the population of people hospitalized with BHD, results indicate that Non-Hispanic Blacks had a higher risk of concurrent TRI compared with Whites hospitalized with BHD, and these findings persisted in the mental illness sub-analysis. Research on Medicare claims administrative data found that Black Medicare beneficiaries had significantly higher risk of inpatient and outpatient visits for hyperthermia and hypothermia [42]. Non-Hispanic Blacks may be at increased risk of BHD with concurrent TRI due to demographic disparities in the social determinants of health, such as differential environmental conditions including housing and neighborhood characteristics and access to heat and air-conditioning [48,49]. Non-Hispanic Blacks experience disparities in access and utilization of health care services and mental health medications, resulting in higher rates of inpatient care for mental health services [50]. Non-Hispanic Blacks with mental illness are more likely to be prescribed anti-psychotics in general, as well as at higher doses [51]. Also, Non-Hispanic Blacks are disproportionately affected by homelessness and therefore exposure to extremes of temperature [52].

### 4.4. Insurance/Access to Care

We observed that among BHD hospitalizations, being on Medicare, Medicaid, and uninsured status were risk factors for concurrent CRI compared to BHD hospitalized patients with private insurance, and this risk persisted in sub-analyses of both mental and substance-use disorders. Patients hospitalized with substance-use disorders had an increased risk of concurrent HRI if they were uninsured or had an “other“ insurance status compared to those with private insurance. Medicare recipients who are 65 years and over are therefore a vulnerable population for CRI. Patients with low socioeconomic status are more likely to be uninsured or on Medicaid and homeless people have a low prevalence of insurance coverage due to barriers to healthcare access [53,54,55]. These vulnerable populations have decreased access to healthcare, resources, and potentially shelter, heating, and air-conditioning. Although we were not able to evaluate homelessness in our study, BHD are prevalent in homeless populations where barriers to healthcare and resource access, particularly shelter and air conditioning during extreme temperature events, are common [56,57]. These factors render the homeless population extremely vulnerable to TRIs, underlining the importance of public health responses when extreme temperatures are expected, especially in extreme cold events where mortality is even greater [56]. For example, a study conducted in New York City observed that approximately one-fourth of those hospitalized for CRI were homeless, and over one-third of those who died from CRI were homeless [20]. In our study population, several diagnoses had decreased risks of hospitalization with a concurrent TRI, such as depressive disorder, bipolar disorders, and anxiety disorders. This decreased risk may possibly be attributed to individuals with these BHD being more likely to stay indoors and not be exposed to extreme temperatures from factors such as impaired decision making, communication, and comprehension. Further research is needed to explore these associations.

### 4.5. Hospital Size & Location

In the study population, concurrent HRI hospitalizations were most likely to be in medium size hospitals compared to large hospitals, while concurrent CRI hospitalization was more likely in rural hospital locations, especially small rural towns, compared to urban BHD hospitalizations. Schmeltz and Gamble similarly observed that BHD patients in medium sized hospitals in the south and midwestern United States had higher risk of hospitalization with concurrent HRI [11]. Schemltz and Gamble also found a higher risk of concurrent HRI in rurally located hospitals; however, in our study, hospital location did not play a significant role in the risk of concurrent HRI among BHD cases. Further analysis could help determine the factors that may contribute to an increased risk of HRI associated with medium sized hospitals. Potential contributary factors to increased CRI in rural areas may be a decreased level of access to health resources and shelter, in addition to a greater likelihood of engagement in outdoor activities. Additionally, in NYS, most rural areas are upstate, which is much colder than downstate.

### 4.6. Strengths & Limitations

Many previous studies focus on the association between temperature and BHD morbidity and mortality, but to our knowledge this is the first study to assess risk factors in BHD hospitalizations for both concurrent heat- and cold-related illnesses in NYS. An important strength of this study is the use of administrative data specific to NYS to perform an in-depth exploration of several BHD. This allowed us to make NYS-specific individual-level analyses and inferences using a large sample size. Cross sectional research such as this is also useful for establishing preliminary findings and creating a direction for future research.

This study has some limitations. Firstly, we are unable to attribute any adverse temperature-related health outcome to specific heat/cold events. We used administrative billing data to define temperature-related hospitalizations. As such, we may only have captured cases severe enough to need hospitalization. Due to differential data entry from provider to provider, there is also the potential for misclassification. In addition, the underreporting of secondary diagnoses can lead to misclassification, which may result in the observed paradoxical decreased risk from comorbidities that are known to increase the risk of TRI [58,59]. We did not have a reliable indicator of homelessness in the SPARCS dataset. Without a clear indication of homelessness in the study population, we had to make assumptions based on the known prevalence of mental health disorders and substance-use issues in this population. An additional limitation, not specific to our study, is the change in ICD-9 to ICD-10 classification systems that led to differential diagnostic coding, particularly with substance-use disorders, and schizophrenia and other psychotic disorders. From the results of a log-binomial regression sensitivity analysis for substance-use disorders and schizophrenia and other psychotic disorders, we observed that the direction of associations for the risk factors were not affected before and after 2015, when the shift from ICD-9 to ICD-10 coding in SPARCS occurred (results not shown) indicating that the difference in classification did not affect our risk factor analyses. Finally, 18% of CRI cases occurred outside of the selected cold months of October–May. We limited the analysis to season periods to avoid overlap but recommend that future studies use data for the whole year to capture all cases.

## 5. Conclusions

Our results agree with prior risk factor findings for BHD hospitalization with a concurrent HRI. This study adds to the literature by identifying dementia, schizophrenia, substance-use disorders, including alcohol dependence and non-dependent substance-use, increasing age, male gender, non-Hispanic Black and other race, Medicare, Medicaid, uninsured status, and respiratory disease as risk factors for BHD hospitalization with a concurrent CRI. Study results can be used to aid public health initiatives from state and local health departments and the community to develop TRI adaptation strategies tailored to the needs of the most vulnerable populations. TRIs are preventable health events that can benefit from targeted programs aimed at preparing the elderly for extreme temperature events, providing shelter for homeless populations, and assisting vulnerable groups with information and resources during extreme temperature events.

## Figures and Tables

**Table 1 ijerph-19-16411-t001:** Patient and hospital characteristics for all hospitalizations with behavioral health disorders (BHD), BHD and heat-related illnesses (HRI) (May-September), and BHD with cold-related illnesses (CRI) (October–April), 2005–2019.

	BHD	BHD and HRI ^1^	BHD and CRI ^2^
**Total N**	12,272,393	1914	7738
**Age in years, Mean (SD)**	56 (20.9)	61.3 (19.5)	58.9 (20.0)
**Age Categories in years, %**			
5–17	309,307 (2.5)	24 (1.3)	113 (1.5)
18–39	2,481,211 (20.2)	269 (14.1)	1205 (15.6)
40–64	5,070,216 (41.3)	758 (39.6)	3517 (45.5)
65–74	1,579,414 (12.9)	303 (15.8)	988 (12.8)
75+	2,832,245 (23.1)	560 (29.3)	1915 (24.8)
**Sex, %**			
Male	6,063,067 (49.4)	1213 (63.4)	5143 (66.5)
Female	6,209,140 (50.6)	701 (36.6)	2595 (33.5)
Missing	186 (.002)	-	-
**Race, %**			
Non-Hispanic White	6,934,606 (56.6)	1015 (53.2)	4157 (53.8)
Hispanic/Latino	1,454,637 (11.9)	229 (12.0)	743 (9.6)
Non-Hispanic Black	2,303,695 (18.8)	417 (21.9)	1799 (23.3)
Asian/Pacific Islander	200,237 (1.6)	31 (1.6)	107 (1.4)
Other ^3^	1,352,740 (11.1)	216 (11.3)	918 (11.9)
Missing	26,478 (0.2)	6 (0.3)	14 (0.1)
**Payer, %**			
Medicare	4,040,182 (34.7)	717 (39.3)	2644 (36.0)
Medicaid	4,341,858 (37.2)	632 (34.6)	3129 (42.6)
Private/HMO	2,258,459 (19.4)	298 (16.3)	879 (11.9)
Uninsured	894,255 (7.7)	155 (8.5)	645 (8.8)
Other ^4^	124,859 (1.1)	24 (1.3)	48 (0.7)
Missing	612,780 (5.0)	88 (4.6)	393 (5.1)
**Comorbidities ^5^, %**			
Diabetes	2,763,430 (22.5)	435 (22.7)	1562 (20.2)
Cardiovascular Disease	2,706,675 (22.1)	637 (22.8)	1406 (18.2)
Respiratory Disease	4,328,933 (35.3)	604 (31.6)	2886 (37.3)
Obesity	1,239,795 (10.1)	157 (8.2)	343 (4.4)
**Hospital Size ^6^, %**			
Small	2,390,623 (19.5)	347 (18.1)	1390 (18.0)
Medium	3,261,862 (26.6)	586 (30.6)	2100 (27.1)
Large	6,619,908 (53.9)	981 (51.3)	4248 (54.9)
**Hospital Location ^7^, %**			
Urban	11,205,295 (91.8)	1769 (92.4)	6980 (90.2)
Large Rural City/Town	722,465 (5.9)	98 (5.1)	504 (6.5)
Small Rural Town	282,392 (2.3)	47 (2.5)	254 (3.2)
Missing	62,241 (0.5)	-	-

^1^ Heat-Related Illnesses (HRI) ICD-9 992, E900.0, E900.9; ICD-10, T67, X30, X32. ^2^ Cold-Related Illnesses (CRI) ICD-9 991, E901.0, E901.9, E988.3; ICD-10 T68, T69, X31, T33, T34. ^3^ Other Race includes Native American, Non-Hispanic multiracial, and other races. ^4^ Other includes Workers’ Compensation, CHAMPUS, Title V, other government programs, and non-government programs.^5^ Comorbidities do not add up to 100%; they are not mutually exclusive and not all patients have comorbidities. ^6^ Hospital Size Small: Rural 1–49 beds, Urban Non-Teaching 1–124 beds, Urban Teaching 1–249 beds; Medium: Rural 50–99 beds, Urban Non-Teaching 125–199 beds, Urban Teaching 250–424 beds; Large: Rural 100+ beds, Urban Non-Teaching 200+ beds, Urban Teaching 425+ beds. https://www.ncbi.nlm.nih.gov/books/NBK373736/table/sb205.t3/ (accessed on 14 December 2021). ^7^ Rural-urban commuting area (RUCA) codes were used to categorize rural and urban areas of NYS.

**Table 2 ijerph-19-16411-t002:** Diagnosis counts and risk ratios (RRs) of hospitalizations due to a primary or secondary diagnosis of behavioral health disorders (BHD) with a concurrent heat-related illness (HRI) (May–September) or cold-related illness (CRI) (October–April), 2005–2019.

BHD ^1^	HRI ^2^				CRI ^3^			
Description	Count	%	RR ^4^	95% CI	Count	%	RR ^4^	95% CI
**Behavioral Health Disorder Diagnoses**	**2661**	**100**			**11,449**	**100**		
Dementia	471	17.7	1.65	(1.49, 1.83)	1882	16.4	1.52	(1.44, 1.60)
Depressive Disorders	489	18.4	0.63	(0.57, 0.70)	1720	15.0	0.53	(0.50, 0.56)
Schizophrenia Spectrum and Other Psychotic Disorders	354	13.3	1.24	(1.10, 1.39)	1401	12.2	1.22	(1.15, 1.30)
*Schizophrenia*	199	56.2	1.38	(1.19, 1.60)	810	57.8	1.41	(1.31, 1.52)
Bipolar and Related Disorders	121	4.5	0.77	(0.64, 0.93)	467	4.1	0.76	(0.69, 0.83)
Anxiety Disorders	215	8.1	0.53	(0.46, 0.61)	745	6.5	0.45	(0.41, 0.48)
Trauma and Stressor-Related Disorders	51	1.9	0.66	(0.50, 0.87)	225	2.0	0.73	(0.64, 0.83)
Attention Deficit Hyperactivity Disorder	25	0.9	0.81	(0.54, 1.20)	79	0.7	0.62	(0.50, 0.78)
Disruptive, Impulse Control, and Conduct Disorders	10	0.4	0.32	(0.17, 0.60)	104	0.9	0.84	(0.70, 1.02)
Personality Disorders	11	0.4	0.19	(0.10, 0.34)	147	1.3	0.63	(0.54, 0.74)
Obsessive Compulsive and Related Disorders	14	0.5	0.82	(0.49, 1.39)	76	0.7	1.16	(0.93, 1.46)
Substance-Use Disorders	900	33.8	0.93	(0.85, 1.02)	4603	40.2	1.62	(1.55, 1.70)
*Alcohol Dependence*	137	15.2	0.61	(0.51, 0.72)	1552	33.7	2.10	(1.99, 2.22)
*Drug Dependence*	107	11.9	0.50	(0.41, 0.61)	434	9.4	0.52	(0.47, 0.57)
*Non-Dependent Drug/Alcohol Use*	632	70.2	1.10	(1.00, 1.21)	2582	56.1	1.20	(1.15, 1.26)

^1^ ICD codes: See Crosswalk in Appendix A. ^2^ Heat-Related Illnesses (HRI) ICD-9 992, E900.0, E900.9; ICD-10, T67, X30, X32. ^3^ Cold-Related Illnesses (CRI) ICD-9 991, E901.0, E901.9, E988.3; ICD-10 T68, T69, X31, T33, T34. ^4^ Risk ratios (relative risk) compared hospitalizations for BHD with concurrent TRI to hospitalizations for BHD without TRI, 2005–2019. The italics is category subsets.

**Table 3 ijerph-19-16411-t003:** Multivariable models of risk factors for hospitalization for all behavioral health disorders, mental illness, and substance-use disorders with a concurrent heat-related illness (May–September), 2005–2019.

Characteristics	All Behavioral Health Disorders		Mental Illness ^1^		Substance-Use Disorders	
	RR	95% CI	RR	95% CI	RR	95% CI
**Age Categories**						
5–17	0.85	(0.56, 1.28)	1.16	(0.66, 2.03)	1.45	(0.68, 3.07)
18–39	1.00^6^	-	1.00	-	1.00	-
40–64	1.46	(1.27, 1.69)	2.27	(1.80, 2.87)	1.25	(1.05, 1.48)
65–74	2.26	(1.87, 2.73)	3.49	(2.63, 4.63)	1.63	(1.23, 2.16)
75+	2.62	(2.19, 3.13)	3.38	(2.56, 4.46)	2.03	(1.45, 2.83)
**Sex**						
Male	1.87	(1.69, 2.06)	1.71	(1.50, 1.96)	2.66	(2.23, 3.17)
Female	1.00	-	1.00	-	1.00	-
**Race**						
Non-Hispanic White	1.00	-	1.00	-	1.00	-
Hispanic/Latino	1.14	(0.98, 1.33)	1.03	(0.82, 1.29)	1.18	(0.96, 1.45)
Non-Hispanic Black	1.34	(1.18, 1.51)	1.30	(1.09, 1.56)	1.17	(0.98, 1.40)
Asian/Pacific Islander	1.10	(0.77, 1.59)	1.16	(0.68, 1.98)	1.43	(0.85, 2.41)
Other ^2^	1.08	(0.92, 1.26)	0.92	(0.73, 1.17)	1.06	(0.85, 1.32)
**Payer**						
Medicare	0.97	(0.83, 1.13)	1.10	(0.88, 1.37)	0.94	(0.74, 1.20)
Medicaid	0.98	(0.85, 1.13)	1.10	(0.90, 1.36)	0.99	(0.82, 1.19)
Private/HMO	1.00	-	1.00	-	1.00	-
Uninsured	1.22	(1.00 1.48)	0.91	(0.65, 1.27)	1.45	(1.13, 1.84)
Other ^3^	1.30	(0.86, 1.97)	0.61	(0.25, 1.50)	1.95	(1.24, 3.08)
**Comorbidities**						
Diabetes	0.91	(0.82, 1.03)	1.05	(0.90. 1.23)	0.91	(0.76, 1.09)
Cardiovascular Disease	0.83	(0.74, 0.93)	0.89	(0.75, 1.05)	0.80	(0.66, 0.97)
Respiratory Disease	0.86	(0.78. 0.95)	0.90	(0.78, 1.04)	1.27	(1.10, 1.47)
Obesity	1.00	(0.84, 1.18)	1.00	(0.80, 1.24)	1.03	(0.80, 1.33)
**Hospital Size ^4^**						
Small	0.98	(0.86, 1.12)	0.90	(0.75, 1.07)	0.82	(0.67, 1.01)
Medium	1.18	(1.07, 1.32)	1.00	(0.86, 1.17)	1.16	(1.00, 1.35)
Large	1.00	-	1.00	-	1.00	-
**Hospital Location ^5^**						
Urban	1.00	-	1.00	-	1.00	-
Large Rural City/Town	0.95	(0.77, 1.18)	0.84	(0.62, 1.13)	1.03	(0.76, 1.39)
Small Rural Town	1.07	(0.79, 1.45)	1.06	(0.69, 1.64)	1.07	(0.68, 1.68)

^1^ Includes depressive disorders, schizophrenia and other psychotic disorders, bipolar and related disorders, anxiety disorders, trauma and stressor related disorders, attention deficit hyperactivity disorders, disruptive, impulse control, and conduct disorders, personality disorders, and obsessive compulsive and related disorders. Excludes dementia since it is not a mental disorder. ^2^ Other Race includes Native American, Non-Hispanic multiracial and other races. ^3^ Other includes Workers’ Compensation, CHAMPUS, Title V, other government programs, and non-government programs. ^4^ Hospital Size Small: Rural 1–49 beds, Urban Non-Teaching 1–124 beds, Urban Teaching 1–249 beds; Medium: Rural 50–99 beds, Urban Non-Teaching 125–199 beds, Urban Teaching 250–424 beds; Large: Rural 100+ beds, Urban Non-Teaching 200+ beds, Urban Teaching 425+ beds. https://www.ncbi.nlm.nih.gov/books/NBK373736/table/sb205.t3/ (accessed on 14 December 2021). ^5^ Rural-urban commuting area (RUCA) codes were used to categorize rural and urban areas of NYS. ^6^ Risk Ratio (RR) of 1.00 indicates reference category.

**Table 4 ijerph-19-16411-t004:** Multivariable model of risk factors for hospitalization for all behavioral health disorders, mental illness, and substance-use disorders with a concurrent cold-related illness (October–April), 2005–2019.

Characteristics	All Behavioral Health Disorders		Mental Illness ^1^		Substance-Use Disorders	
	RR	95% CI	RR	95% CI	RR	95% CI
**Age Categories**						
5–17	0.76	(0.62, 0.92)	0.53	(0.40, 0.72)	2.34	(1.79, 3.07)
18–39	1.00^6^	-	1.00	-	1.00	-
40–64	1.53	(1.43, 1.64)	1.48	(1.34, 1.63)	1.48	(1.37, 1.60)
65–74	1.55	(1.40, 1.71)	1.24	(1.08, 1.43)	1.68	(1.49, 1.90)
75+	1.78	(1.63, 1.95)	1.34	(1.17, 1.53)	1.64	(1.40, 1.92)
**Sex**						
Male	2.06	(1.96, 2.17)	1.89	(1.76, 2.03)	2.29	(2.13, 2.47)
Female	1.00	-	1.00	-	1.00	-
**Race**						
Non-Hispanic White	1.00	-	1.00	-	1.00	-
Hispanic/Latino	0.80	(0.74, 0.87)	0.60	(0.52, 0.70)	0.74	(0.67, 0.82)
Non-Hispanic Black	1.23	(1.15, 1.30)	1.28	(1.17, 1.41)	0.91	(0.84, 0.98)
Asian/Pacific Islander	0.80	(0.65, 0.98)	0.72	(0.51, 1.01)	0.95	(0.73, 1.23)
Other ^2^	1.07	(0.99, 1.15)	0.91	(0.81, 1.03)	1.03	(0.94, 1.14)
**Payer**						
Medicare	1.51	(1.38, 1.64)	1.50	(1.33, 1.70)	1.43	(1.27, 1.60)
Medicaid	1.71	(1.59, 1.85)	1.57	(1.40, 1.75)	1.66	(1.52, 1.82)
Private/HMO	1.00	-	1.00	-	1.00	-
Uninsured	1.76	(1.59, 1.94)	1.45	(1.24, 1.71)	1.78	(1.58, 2.00)
Other ^3^	0.84	(0.63, 1.12)	0.75	(0.48, 1.16)	0.78	(0.54, 1.11)
**Comorbidities**						
Diabetes	0.89	(0.84, 0.95)	0.97	(0.89, 1.06)	0.91	(0.83, 0.99)
Cardiovascular Disease	0.68	(0.64, 0.72)	0.72	(0.65, 0.79)	0.60	(0.54, 0.66)
Respiratory Disease	1.08	(1.03, 1.14)	1.14	(1.06, 1.22)	1.18	(1.11, 1.26)
Obesity	0.49	(0.43, 0.54)	0.50	(0.43, 0.58)	0.47	(0.40, 0.55)
**Hospital Size ^4^**						
Small	0.92	(0.86, 0.98)	0.94	(0.86, 1.03)	0.78	(0.72, 0.84)
Medium	0.95	(0.90, 1.01)	0.93	(0.85, 1.01)	0.86	(0.80, 0.92)
Large	1.00	-	1.00	-	1.00	-
**Hospital Location ^5^**						
Urban	1.00	-	1.00	-	1.00	-
Large Rural City/Town	1.24	(1.13, 1.36)	1.25	(1.10, 1.43)	1.00	(0.88, 1.14)
Small Rural Town	1.54	(1.35, 1.75)	1.58	(1.30, 1.91)	1.05	(0.86, 1.28)

^1^ Includes depressive disorders, schizophrenia and other psychotic disorders, bipolar and related disorders, anxiety disorders, trauma and stressor related disorders, attention deficit hyperactivity disorders, disruptive, impulse control, and conduct disorders, personality disorders, and obsessive compulsive and related disorders. Excludes dementia since it is not a mental disorder. ^2^ Other Race includes Native American, Non-Hispanic multiracial and other races. ^3^ Other includes Workers’ Compensation, CHAMPUS, Title V, other government programs, and non-government programs. ^4^ Hospital Size Small: Rural 1–49 beds, Urban Non-Teaching 1–124 beds, Urban Teaching 1–249 beds; Medium: Rural 50–99 beds, Urban Non-Teaching 125–199 beds, Urban Teaching 250–424 beds; Large: Rural 100+ beds, Urban Non-Teaching 200+ beds, Urban Teaching 425+ beds. https://www.ncbi.nlm.nih.gov/books/NBK373736/table/sb205.t3/ (accessed on 14 December 2021). ^5^ Rural-urban commuting area (RUCA) codes were used to categorize rural and urban areas of NYS. ^6^ Risk Ratio (RR) of 1.00 indicates reference category.

## Data Availability

Data for this study came from the NYS Statewide Planning and Research Cooperative System (SPARCS) database. This data is not publicly available as it would compromise individual privacy.

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
