# Peer review of "Identifying Risk Factors for Hospitalization with Behavioral Health Disorders and Concurrent Temperature-Related Illness in New York State"

_ijerph, 2022, doi:10.3390/ijerph192416411_

Round 1

Reviewer 1 Report

A very interesting study.  Not being from the US, it would be useful to have a bit more context on NYS included, such as the number of hospitals in each classification.  The comment regarding the location of "most rural areas are upstate" is useful and more detail could add to the interpretation of the results.

What proportion of overall hospitalisations during this timeframe does the 12 million BHD presentations represent?  This will again provide some context of BHD overall.  Proportions of patients hospitalised with BHD and TRI should also be commented on - it appears that there was an average of less than one patient per hospital per year for BHD and HRI and drawing conclusions from this low number is difficult.  It is still an important issue to note, but there could be other factors that are not evident with these numbers over the timeframe and area.  This should also be included in the limitations.

Clarification of the "18% of CRI cases were not captured between October-may" (#490) is needed.  Does this mean of all CRI cases for the year 18% occurred outside of October-May (which I think it does), or that between October and May, 18% of cases were not captured for some reason?

There are a few minor grammatical errors throughout (including #58,#443 ?words missing) but otherwise reads very well.

Reviewer 2 Report

1.     Line 34, change “on” to “an”

2.     Line 58, change “HRI” to “of HRI”

3.     Line 114, full name of ICD

4.     Table 3 and 4, what does “ref.” mean?

5.     Appendix D, change MBD to BHD in figure (a) and (b) to keep it consistent,  or explain the difference between MBD and BHD in the figure legend. 

Reviewer 3 Report

Manuscript ID ijerph-2039302 are dealing with important and topic related to the specific-causes hospitalisation, considering heat and cold-related illness. Still, manuscript should be strongly revised due to several reasons:

Methodological approach in the manuscript is not completely clear and understandable. Authors used TRI (temperature-related illness), but it is not clear how they reach this outcome? Which indicator from climate was used(e.g. Ta, Tmin, Tmax, some other heat index….)?

Simultaneously, manuscript is closely related with heat and cold-related illness, but in paper are missing data about climate characteristics in NYS for 2005-2019. 

Methodological chapter, as well as results should be extended with missing data about temperature. Furthermore, chapter Discussion should be improved with observation of temperature indicator/s.

Statistical analysis, related to the presented model did not include any climate indicator. Please, explain why?

Round 2

Reviewer 3 Report

Revised version of manuscript ID ijerph-2039302 has improved according to the suggestions. Additionally, for comment related to the “Which indicator from climate was used…” authors clarified methodology approach regarding to the ICD codes.

Authors, also adopted review about climate indicators and they are presented in Supplementary file (Appendix F, Table A5), but this appendix was not included in manuscript) it was not cited in main text). Furthermore, short explanation of presented data in table A5 should be included in results.
